# Correlation of ammonia and blood laboratory parameters with hepatic encephalopathy: A systematic review and meta-analysis

Ali Sepehrinezhad[1,2], Negin Ghiyasi Moghaddam[1], Navidreza Shayan[1], Sajad Sahab Negah [1,2,3]*

1 Neuroscience Research Center, Mashhad University of Medical Sciences, Mashhad, Iran, 2 Department of Neuroscience, Faculty of Medicine, Mashhad University of Medical Sciences, Mashhad, Iran, 3 Shefa Neuroscience Research Center, Khatam Alanbia Hospital, Tehran, Iran

* sahabnegahs@mums.ac.ir

## Abstract

### Background and objectives

Emerging research suggests that hyperammonemia may enhance the probability of hepatic encephalopathy (HE), a condition associated with elevated levels of circulating ammonia in patients with cirrhosis. However, some studies indicate that blood ammonia levels may not consistently correlate with the severity of HE, highlighting the complex pathophysiology of this condition.

### Methods

A systematic review and meta-analysis through PubMed, Scopus, Embase, Web of Science, and Virtual Health Library were conducted to address this complexity, analyzing and comparing published data on various laboratory parameters, including circulating ammonia, blood creatinine, albumin, sodium, and inflammation markers in cirrhotic patients, both with and without HE.

### Results

This comprehensive review, which included 81 studies from five reputable databases until June 2024, revealed a significant increase in circulating ammonia levels in cirrhotic patients with HE, particularly those with overt HE. Notably, significant alterations were observed in the circulating creatinine, albumin, sodium, interleukin-6 (IL-6), and tumor necrosis factor-alpha (TNFα) in HE patients.

### Conclusions

These findings suggest an association between ammonia and HE and underscore the importance of considering other blood parameters such as creatinine, albumin, sodium, and pro-inflammatory cytokines when devising new treatment strategies for HE.

**Data Availability Statement:** All relevant data are within the manuscript and its Supporting Information files.

**Funding:** The author(s) received no specific funding for this work.

**Competing interests:** The authors have declared that no competing interests exist

## Introduction

Hepatic encephalopathy (HE) is the main complication of advanced liver disease and portosystemic shunt that is characterized by the development of the broad spectrum of neurological and neuropsychiatric disturbances from minimal changes in cognitive performances in covert HE (CHE) to gross disorientation and motor system abnormalities in overt HE (OHE) [1–3]. HE is associated with lower patient quality of life, increasing disabilities, being the primary cause of ER hospitalization, and showing a poor prognosis [4, 5]. In patients with cirrhosis, the prevalence of CHE ranges from 20% to 80%; nevertheless, 40% of these patients experienced OHE [6–8].

Although the majority of studies agree that ammonia and inflammation are key factors in the pathophysiology of HE, the specific underlying mechanisms of HE remain unclear [9, 10]. It has been demonstrated that cirrhotic individuals with HE, had elevated blood levels of ammonia, often referred to as hyperammonemia conditions [9, 11, 12]. This condition increased cerebral uptake of ammonia in HE patients [13–15]. Ammonia, a byproduct produced by gut microbes during the breakdown of nitrogen-containing compounds, is detoxified by intact hepatocytes through the production of urea under physiological conditions [16]. Numerous studies suggest that hyperammonemia may predispose patients with cirrhosis to HE, and it is associated with the severity of HE [17–22]. However, conflicting findings have emerged from multiple studies, suggesting that circulating ammonia may not be a suitable marker for evaluating HE in cirrhotic patients [23–25]. It has been proposed that systemic inflammation and ammonia might synergistically promote the progression of HE following liver diseases [25–29]. In contrast, a research team found that systemic inflammation alone did not correlate with the development of HE [30] or cognitive impairments [31] in cirrhotic patients, and anti-inflammatory therapy did not improve cognitive deficits in HE rats [32]. Several studies have also identified INR, white blood cells, hyponatremia, bilirubin, and blood creatinine as potential risk factors for HE [33–36]. Despite recent investigations, the relationship between these blood parameters and HE remains a topic of ongoing discussion. Therefore, we conducted a comprehensive review and meta-analysis to elucidate any potential correlations between circulating ammonia levels, inflammation, and several laboratory parameters with HE, integrating and examining all available results.

## Methods

### Data sources and search

We conducted a systematic review and meta-analysis based on the Preferred Reporting Items for Systematic Reviews and Meta-Analyses (PRISMA) standards. The literature review and search of PubMed, Scopus, Embase, Web of Science, and Virtual Health Library (VHL) were used to find all original studies until June 2024. Three investigators used the following Medical Subject Headings (MESH) terms in this study: "hepatic encephalopathy", "hepatic coma", "portal systemic encephalopathy", "hepatocerebral encephalopathy", "portosystemic encephalopathy", "ammonia", "hyperammonemia", and "hyperammonemic". A combination of these terms was investigated using an advanced search in the aforementioned databases (S2 Table in S1 File).

### Study selection, data extraction and quality assessment

All extracted papers were exported into an Excel file and classified according to some properties and parameters including title, authors, publication year, age, country, ammonia, albumin, platelets, total bilirubin, alanine aminotransferase (ALT), aspartate aminotransferase (AST), gamma-glutamyl transpeptidase (GGT), creatinine, hemoglobin, prothrombin time (PT),

International Normalized Ratio (INR), sodium, white blood cell, Model for End-stage Liver Disease (MELD) score, and Child-Pugh score. Before the quality assessment, exported papers underwent two rounds of screening. Three investigators reviewed each title and abstract in Step 1 to make sure they met the inclusion criteria. At the subsequent stages, all investigators carefully evaluate the full text of extracted studies according to the following criteria. Following strict quality control (appraisal check), all case-control and cross-sectional studies reporting cirrhosis patients (of any etiology) with HE was included in this analysis. The following criteria must be met by all included studies: I. Cirrhosis and HE were determined according to a valid and reliable diagnosis method; II. Patients should not have undergone liver transplantation during the study; III. The study reported mean and SD for quantitative parameters; IV. The study should report the average levels of ammonia in the cirrhosis group and HE, V. The levels of ammonia in case and control groups should be measured by the same and reliable method. Case reports, correspondence, review papers, in vitro studies, animal studies, randomized controlled trials or interventional studies, letters, books, conference papers, and editorials were not included in this study. The Joanna Briggs Institute (JBI) critical appraisal checklist for case-control studies was used to assess the risk of bias. Papers with a score of 5–10 were considered for meta-analysis as high-quality papers.

## Data analysis

All analyses were conducted by the Review Manager (RevMan) software ver. 5 (Copenhagen: The Nordic Cochrane Centre, The Cochrane Collaboration, 2008). The standardized mean differences for ammonia and other parameters were calculated and a random-effects analysis model was applied. Moreover, heterogeneity between studies was assessed using $I^2$ criterion ($I^2 \geq 75\%$ specified substantial heterogeneity). Furthermore, a p-value of less than 0.05 is used as a statistical significance cutoff. We also performed a subgroup analysis based on HE types (MHE, CHE, and OHE) and the level of circulatory ammonia.

## Results

A total of 25121 papers were included in our literature review (Fig 1). Following the application of our intended criteria and full-text screening, we selected 81 high-quality papers for final analysis (Table 1 and S3 Table in S1 File).

The studies included two groups of patients: a control group (cirrhosis patients without HE) and a case group (cirrhosis patients with HE). Sixty-three publications were suitable for determining the standardized mean difference in ammonia between HE (N = 1771) and control (N = 2558) groups (Fig 2). We compared the circulatory levels of ammonia in the HE groups—which included all HE types, including MHE, CHE, and OHE—to the control group in these analyses. The mean circulating ammonia in the cirrhosis with HE group was 127.676μg/dl compared to 92.503μg/dl in cirrhosis without HE individuals. There was high heterogeneity in this analysis; however, the forest plot of the included studies using random-effect analysis showed a significant increase in the mean difference of ammonia levels in the HE groups compared to the control (P < 0.00001; $I^2$ = 90%; Fig 2).

Subgroup analysis was then used to compare the ammonia levels between the control group and several types of HE. To compare the ammonia levels between MHE (N = 874) and control (N = 1199) groups, twenty-nine studies were enrolled (Fig 3A). The mean circulating level of ammonia in the cirrhosis with MHE group was 125.19μg/dl compared to 106.482μg/dl in cirrhosis without HE individuals. The standardized mean difference of ammonia was significantly elevated in MHE group compared to cirrhosis-control (P < 0.0001; $I^2$ = 91%; Fig 3A). Moreover, eight papers were included for analyzing ammonia levels between CHE (N = 263)

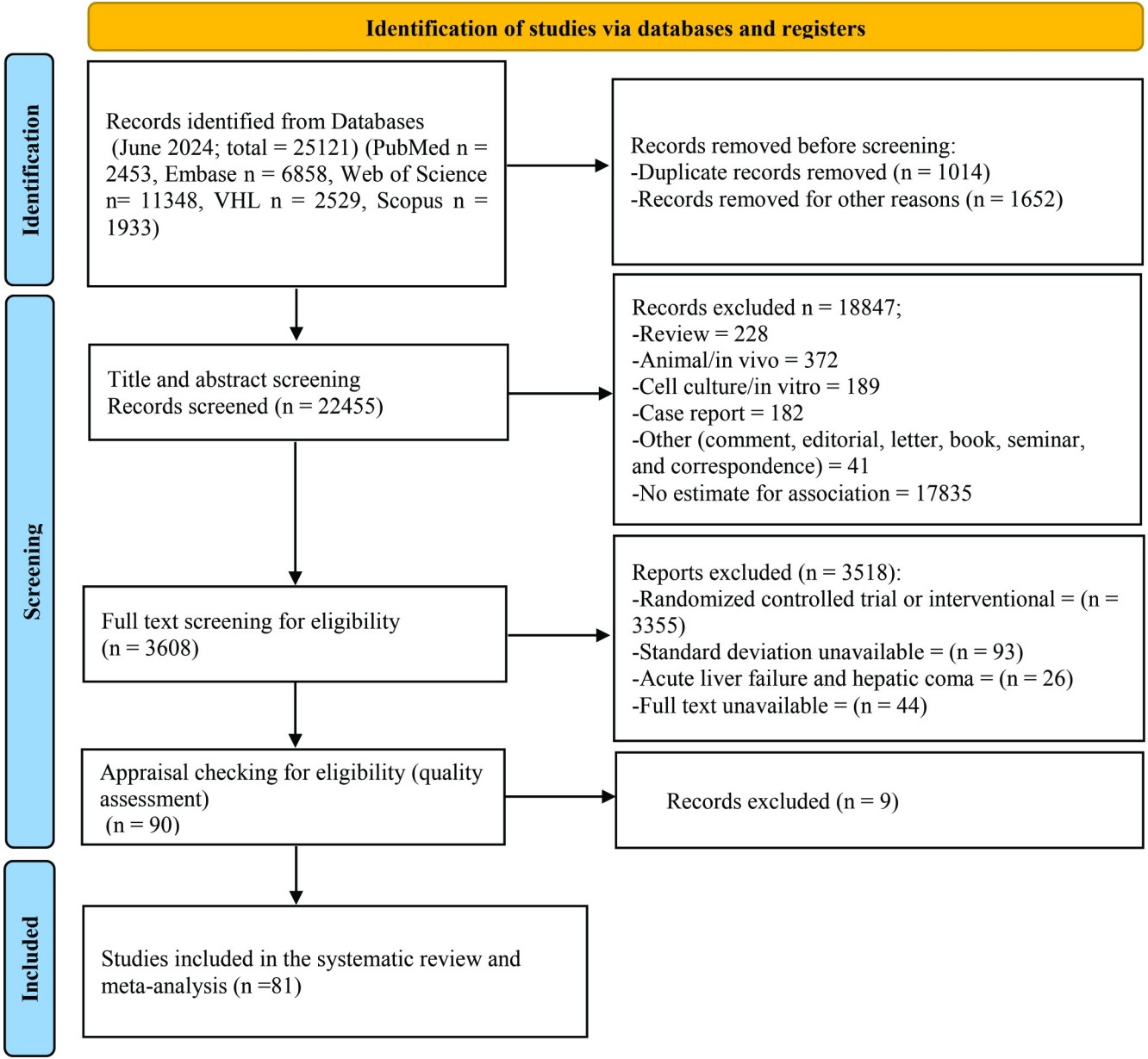

**Fig 1. PRISMA flow chart of the systematic review that represents our literature review process.** PRISMA, Preferred Reporting Items for Systematic Reviews and Meta-Analyses.

and control (N = 559) groups. The mean circulating level of ammonia in the cirrhosis with CHE group was 95.15μg/dl compared to 85.94μg/dl in cirrhosis without HE individuals. The results showed that there was no statistical difference in ammonia levels between the groups ($P = 0.27$; $I^2 = 95\%$; Fig 3B). Twelve articles were subjected to another comparison of average ammonia levels between the control (N = 330) and OHE (N = 280) groups. The mean circulating level of ammonia in the cirrhosis with OHE group was 138.60μg/dl compared to 71.57μg/dl in cirrhosis without HE volunteers. Ammonia levels in the circulation were significantly higher in the OHE group ($P < 0.00001$; $I^2 = 83\%$; Fig 3C).

**Table 1. Characteristics of total included studies for the meta-analysis.**

| Index | Author | Year | Country | Journal | Reference |
|---|---|---|---|---|---|
| 1. | Tran et al. | 2021 | USA | Journal of Neuroimaging | [37] |
| 2. | Ampuero et al. | 2020 | Spain | Liver International | [38] |
| 3. | Tsai et al. | 2019 | Taiwan | Scientific reports | [39] |
| 4. | Nardelli et al. | 2019 | Italy | Hepatology | [40] |
| 5. | Zhang et al. | 2018 | Germany | Korean Journal of Radiology | [41] |
| 6. | Lu et al. | 2018 | China | NeuroReport | [42] |
| 7. | Cheng et al. | 2018 | China | Metabolic Brain Disease | [43] |
| 8. | Zheng et al. | 2017 | China | European Radiology | [19] |
| 9. | Zhang et al. | 2017 | Germany | Brain Imaging and Behavior | [11] |
| 10. | Zhou et al. | 2016 | China | Gastroenterology Research and Practice | [44] |
| 11. | Thomsen et al. | 2016 | UK | PLOS ONE | [45] |
| 12. | Schiff et al. | 2016 | China | Hepatology | [46] |
| 13. | Iwasa et al. | 2016 | Japan | Metabolic Brain Disease | [47] |
| 14. | Rodríguez et al. | 2015 | USA | Liver International | [48] |
| 15. | Jao et al. | 2015 | Taiwan | NeuroImage | [49] |
| 16. | Barbosa et al. | 2015 | Portugal | Acta Médica Portuguesa | [50] |
| 17. | Zheng et al. | 2014 | China | BioMed Research International | [51] |
| 18. | Zhang et al. | 2014 | China | European Journal of Radiology | [52] |
| 19. | Felipo et al. | 2014 | Spain | World Journal of Gastroenterology | [53] |
| 20. | Zhang et al. | 2013 | China | PLOS ONE | [54] |
| 21. | Luo et al. | 2013 | China | Clinics and Research in Hepatology and Gastroenterology | [55] |
| 22. | Felipo et al. | 2013 | Spain | Liver International | [56] |
| 23. | Michalska et al. | 2013 | Poland | Gastroenterology Review | [57] |
| 24. | Ni et al. | 2012 | China | PLOS ONE | [58] |
| 25. | Luo et al. | 2012 | China | Hepatology Research | [59] |
| 26. | Srivastava et al. | 2011 | India | Journal of Gastroenterology and Hepatology | [60] |
| 27. | Gad et al. | 2011 | Egypt | Arab Journal of Gastroenterology | [61] |
| 28. | Sharma et al. | 2010 | India | Saudi Journal of Gastroenterology | [62] |
| 29. | Goel et al. | 2010 | India | Liver International | [63] |
| 30. | Montoliu et al. | 2009 | Spain | Journal of Clinical Gastroenterology | [64] |
| 31. | Montoliu et al. | 2007 | Spain | Journal of Molecular Medicine | [65] |
| 32. | Kundra et al. | 2005 | India | Clinical Biochemistry | [20] |
| 33. | Nicolao et al. | 2003 | Italy | Journal of Hepatology | [66] |
| 34. | Romero-Gómez et al. | 2001 | Spain | American Journal of Gastroenterology | [67] |
| 35. | Testa R et al. | 1989 | Italy | Italian journal of neurological sciences | [68] |
| 36. | McCLAIN, et al. | 1980 | USA | Gut | [69] |
| 37. | Reichert et al. | 2020 | Germany | Digestive Diseases | [70] |
| 38. | Abid et al. | 2020 | Pakistan | Scientific reports | [71] |
| 39. | Zeng et al. | 2019 | China | Journal of Gastroenterology and Hepatology | [72] |
| 40. | Yousif et al. | 2019 | Egypt | Internal and Emergency Medicine | [73] |
| 41. | Yoon et al. | 2019 | Korea | Scientific Reports | [74] |
| 42. | Tan et al. | 2019 | China | British Journal of Biomedical Science | [75] |
| 43. | Sato et al. | 2019 | Japan | Internal Medicine Journal | [76] |
| 44. | Metwally et al. | 2019 | Egypt | European Journal of Gastroenterology & Hepatology | [77] |
| 45. | Li et al. | 2019 | China | Neuroradiology | [78] |
| 46. | Wang et al. | 2017 | China | World Journal of Gastroenterology | [79] |
| 47. | Coskun et al. | 2017 | Turkey | Turkish Journal of Gastroenterology | [80] |

*(Continued)*

**Table 1.** (Continued)

| Index | Author | Year | Country | Journal | Reference |
|---|---|---|---|---|---|
| 48. | Jeong et al. | 2017 | Korea | Journal of Korean Medical Science | [81] |
| 49. | Ruiz-Margáin et al. | 2016 | Mexico | World Journal of Gastroenterology | [82] |
| 50. | Lauridsen et al. | 2016 | USA | Clinical Gastroenterology and Hepatology | [83] |
| 51. | Chen et al. | 2016 | China | Scientific Reports | [84] |
| 52. | Tsai et al. | 2015 | Taiwan | PLOS One | [85] |
| 53. | Riggio et al. | 2015 | Italy | Clinical Gastroenterology and Hepatology | [86] |
| 54. | Wei Li et al. | 2015 | China | Hepatology International | [87] |
| 55. | Jindal et al. | 2015 | India | Digestive and Liver Disease | [88] |
| 56. | Patidar et al. | 2014 | USA | American Journal of Gastroenterology | [89] |
| 57. | Kircheis et al. | 2014 | Germany | Gastroenterology | [90] |
| 58. | Hassan et al. | 2014 | Egypt | Arab Journal of Gastroenterology | [91] |
| 59. | Cona et al. | 2014 | Italy | Clinical Neurophysiology | [92] |
| 60. | Zhang et al.* | 2013 | China | American Journal of Gastroenterology | [93] |
| 61. | Merli et al. | 2013 | Italy | Metabolic Brain Disease | [94] |
| 62. | Li et al. | 2013 | China | World Journal of Gastroenterology | [95] |
| 63. | Sharma et al. | 2012 | India | Saudi Journal of Gastroenterology | [96] |
| 64. | Wunsch et al. | 2011 | Poland | Liver International. | [97] |
| 65. | Riggio et al. | 2011 | Italy | Clinical Gastroenterology and Hepatology | [98] |
| 66. | Duarte-Rojo et al. | 2011 | Mexico | Digestive Diseases and Sciences | [99] |
| 67. | Tan et al. | 2009 | Singapore | Singapore Medical Journal | [100] |
| 68. | Kircheis et al. | 2009 | Germany | Gastroenterology | [101] |
| 69. | Sugimoto et al. | 2008 | Japan | Official journal of the American College of Gastroenterology | [102] |
| 70. | Chakrabarti et al. | 2002 | Italy | Journal of Clinical Gastroenterology | [103] |
| 71. | Alvarez-Leal et al. | 2001 | Mexico | American Journal of Human Biology | [104] |
| 72. | Lee et al. | 1999 | Korea | American Journal of Gastroenterology | [105] |
| 73. | Zheng et al. | 2013 | China | European Journal of Radiology | [106] |
| 74. | Tao et al. | 2013 | China | European Journal of Radiology | [107] |
| 75. | Iversen et al. | 2014 | Denmark | Frontiers in Neuroscience | [108] |
| 76. | Kooka et al. | 2016 | Japan | Hepatology Research | [109] |
| 77. | Garcia-Garcia et al. | 2017 | Spain | PLOS ONE | [110] |
| 78. | Formentin et al. | 2019 | Italy | Journal of Hepatology | [111] |
| 79. | Mangini et al. | 2023 | Italy | Digestive and Liver Disease | [112] |
| 80. | Kapoor et al. | 2023 | India | Turk J Gastroenterol | [113] |
| 81. | Fiorillo et al. | 2023 | Spain | International Journal of Molecular Sciences | [114] |

Our analysis of 31 studies (2700 participants) revealed that patients with cirrhosis and HE had a mean creatinine level of 1.057mg/dl, significantly higher than the 0.939mg/dl observed in cirrhosis patients without HE. ($P < 0.0001$; $I^2 = 48\%$; Fig 4A). Fifty-three papers were used to compare the albumin levels between groups (4627 participants). The circulatory levels of albumin were significantly lower in cirrhotic patients with HE (mean = 3.19g/dl) in comparison to cirrhotic control (mean = 3.59g/dl) group ($P < 0.00001$; $I^2 = 85\%$; Fig 4B). Our meta-analysis of 22 studies (2683 participants) revealed significantly lower blood sodium levels in cirrhotic patients with HE (mean = 135.977 mEq/L) compared to cirrhotic controls without HE (mean = 137.69 mEq/L; $P < 0.00001$). Notably, low heterogeneity was observed across studies for this parameter ($I^2 = 64\%$; Fig 4C).

On the other hand, in 12 enrolled papers (611 participants), blood interleukin-6 (IL-6) levels in cirrhotic patients with HE were higher than those in cirrhotic controls ($P < 0.00001$; $I^2 =$

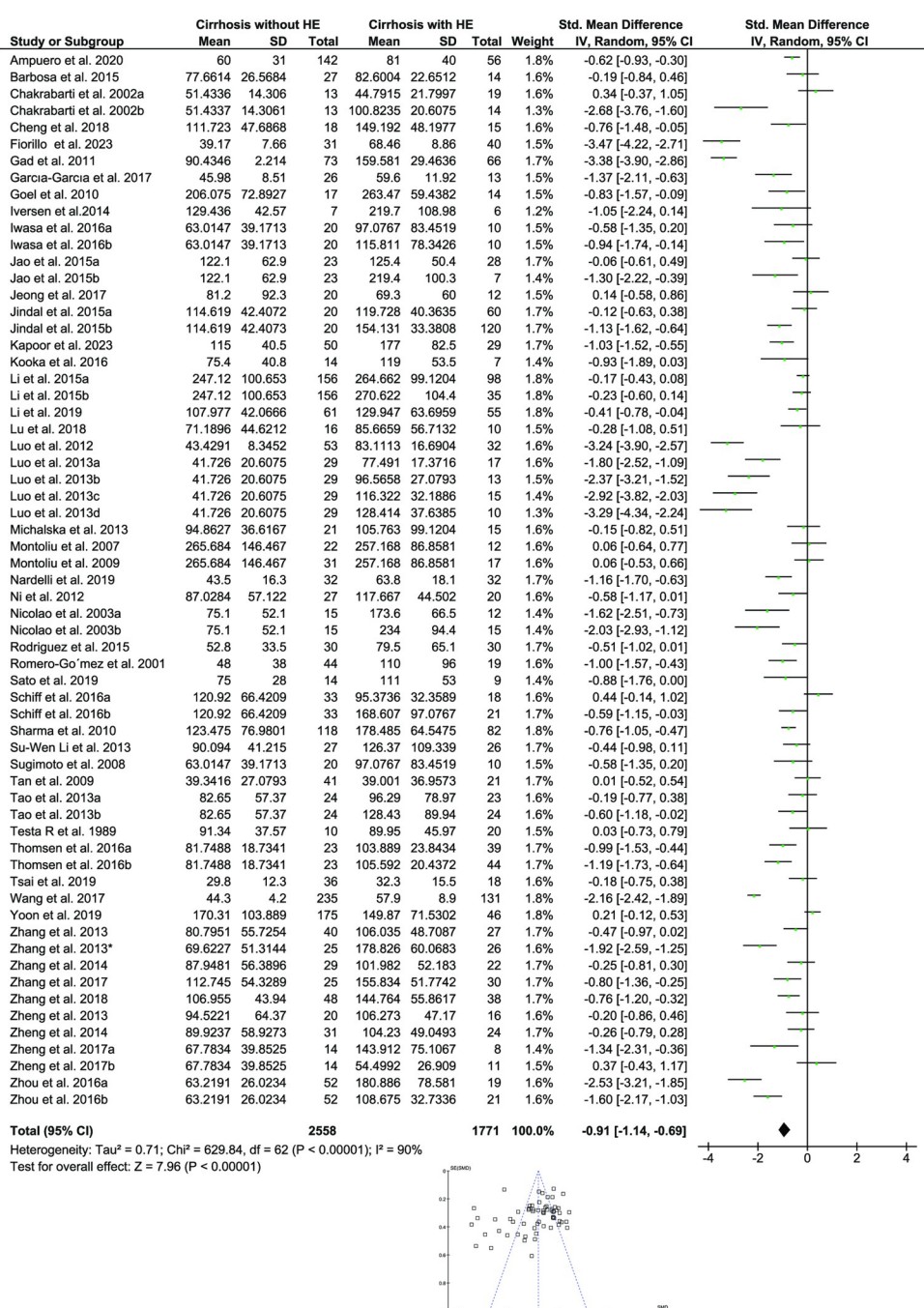

| Study or Subgroup | Cirrhosis without HE | | | Cirrhosis with HE | | | | Std. Mean Difference | Std. Mean Difference |
|---|---|---|---|---|---|---|---|---|---|
| | Mean | SD | Total | Mean | SD | Total | Weight | IV, Random, 95% CI | IV, Random, 95% CI |
| Ampuero et al. 2020 | 60 | 31 | 142 | 81 | 40 | 56 | 1.8% | -0.62 [-0.93, -0.30] | |
| Barbosa et al. 2015 | 77.6614 | 26.5684 | 27 | 82.6004 | 22.6512 | 14 | 1.6% | -0.19 [-0.84, 0.46] | |
| Chakrabarti et al. 2002a | 51.4336 | 14.306 | 13 | 44.7915 | 21.7997 | 19 | 1.6% | 0.34 [-0.37, 1.05] | |
| Chakrabarti et al. 2002b | 51.4337 | 14.3061 | 13 | 100.8235 | 20.6075 | 14 | 1.3% | -2.68 [-3.76, -1.60] | |
| Cheng et al. 2018 | 111.723 | 47.6868 | 18 | 149.192 | 48.1977 | 15 | 1.6% | -0.76 [-1.48, -0.05] | |
| Fiorillo et al. 2023 | 39.17 | 7.66 | 31 | 68.46 | 8.86 | 40 | 1.5% | -3.47 [-4.22, -2.71] | |
| Gad et al. 2011 | 90.4346 | 2.214 | 73 | 159.581 | 29.4636 | 66 | 1.7% | -3.38 [-3.90, -2.86] | |
| García-García et al. 2017 | 45.98 | 8.51 | 26 | 59.6 | 11.92 | 13 | 1.5% | -1.37 [-2.11, -0.63] | |
| Goel et al. 2010 | 206.075 | 72.8927 | 17 | 263.47 | 59.4382 | 14 | 1.5% | -0.83 [-1.57, -0.09] | |
| Iversen et al.2014 | 129.436 | 42.57 | 7 | 219.7 | 108.98 | 6 | 1.2% | -1.05 [-2.24, 0.14] | |
| Iwasa et al. 2016a | 63.0147 | 39.1713 | 20 | 97.0767 | 83.4519 | 10 | 1.5% | -0.58 [-1.35, 0.20] | |
| Iwasa et al. 2016b | 63.0147 | 39.1713 | 20 | 115.811 | 78.3426 | 10 | 1.5% | -0.94 [-1.74, -0.14] | |
| Jao et al. 2015a | 122.1 | 62.9 | 23 | 125.4 | 50.4 | 28 | 1.7% | -0.06 [-0.61, 0.49] | |
| Jao et al. 2015b | 122.1 | 62.9 | 23 | 219.4 | 100.3 | 7 | 1.4% | -1.30 [-2.22, -0.39] | |
| Jeong et al. 2017 | 81.2 | 92.3 | 20 | 69.3 | 60 | 12 | 1.5% | 0.14 [-0.58, 0.86] | |
| Jindal et al. 2015a | 114.619 | 42.4072 | 20 | 119.728 | 40.3635 | 60 | 1.7% | -0.12 [-0.63, 0.38] | |
| Jindal et al. 2015b | 114.619 | 42.4073 | 20 | 154.131 | 33.3808 | 120 | 1.7% | -1.13 [-1.62, -0.64] | |
| Kapoor et al. 2023 | 115 | 40.5 | 50 | 177 | 82.5 | 29 | 1.7% | -1.03 [-1.52, -0.55] | |
| Kooka et al. 2016 | 75.4 | 40.8 | 14 | 119 | 53.5 | 7 | 1.4% | -0.93 [-1.89, 0.03] | |
| Li et al. 2015a | 247.12 | 100.653 | 156 | 264.662 | 99.1204 | 98 | 1.8% | -0.17 [-0.43, 0.08] | |
| Li et al. 2015b | 247.12 | 100.653 | 156 | 270.622 | 104.4 | 33 | 1.8% | -0.23 [-0.60, 0.14] | |
| Li et al. 2019 | 107.977 | 42.0666 | 61 | 129.947 | 63.6959 | 55 | 1.8% | -0.41 [-0.78, -0.04] | |
| Lu et al. 2018 | 71.1896 | 44.6212 | 16 | 85.6659 | 56.7132 | 10 | 1.5% | -0.28 [-1.08, 0.51] | |
| Luo et al. 2012 | 43.4291 | 8.3452 | 53 | 83.1113 | 16.6904 | 32 | 1.6% | -3.24 [-3.90, -2.57] | |
| Luo et al. 2013a | 41.726 | 20.6075 | 29 | 77.491 | 17.3716 | 17 | 1.6% | -1.80 [-2.52, -1.09] | |
| Luo et al. 2013b | 41.726 | 20.6075 | 29 | 96.5658 | 27.0793 | 13 | 1.5% | -2.37 [-3.21, -1.52] | |
| Luo et al. 2013c | 41.726 | 20.6075 | 29 | 116.322 | 32.1886 | 15 | 1.4% | -2.92 [-3.82, -2.03] | |
| Luo et al. 2013d | 41.726 | 20.6075 | 29 | 128.414 | 37.6385 | 10 | 1.3% | -3.29 [-4.34, -2.24] | |
| Michalska et al. 2013 | 94.8627 | 36.6167 | 21 | 105.763 | 99.1204 | 15 | 1.6% | -0.15 [-0.82, 0.51] | |
| Montoliu et al. 2007 | 265.684 | 146.467 | 22 | 257.168 | 86.8581 | 12 | 1.6% | 0.06 [-0.64, 0.77] | |
| Montoliu et al. 2009 | 265.684 | 146.467 | 31 | 257.168 | 86.8581 | 17 | 1.6% | 0.06 [-0.53, 0.66] | |
| Nardelli et al. 2019 | 43.5 | 16.3 | 32 | 63.8 | 18.1 | 32 | 1.7% | -1.16 [-1.70, -0.63] | |
| Ni et al. 2012 | 87.0284 | 57.122 | 27 | 117.667 | 44.502 | 20 | 1.6% | -0.58 [-1.17, 0.01] | |
| Nicolao et al. 2003a | 75.1 | 52.1 | 15 | 173.6 | 66.5 | 12 | 1.4% | -1.62 [-2.51, -0.73] | |
| Nicolao et al. 2003b | 75.1 | 52.1 | 15 | 234 | 94.4 | 15 | 1.4% | -2.03 [-2.93, -1.12] | |
| Rodriguez et al. 2015 | 52.8 | 33.5 | 30 | 79.5 | 65.1 | 30 | 1.7% | -0.51 [-1.02, 0.01] | |
| Romero-Go´mez et al. 2001 | 48 | 38 | 44 | 110 | 96 | 19 | 1.6% | -1.00 [-1.57, -0.43] | |
| Sato et al. 2019 | 75 | 28 | 14 | 111 | 53 | 9 | 1.4% | -0.88 [-1.76, 0.00] | |
| Schiff et al. 2016a | 120.92 | 66.4209 | 33 | 95.3736 | 32.3589 | 18 | 1.6% | 0.44 [-0.14, 1.02] | |
| Schiff et al. 2016b | 120.92 | 66.4209 | 33 | 168.607 | 97.0767 | 21 | 1.7% | -0.59 [-1.15, -0.03] | |
| Sharma et al. 2010 | 123.475 | 76.9801 | 118 | 178.485 | 64.5475 | 82 | 1.8% | -0.76 [-1.05, -0.47] | |
| Su-Wen Li et al. 2013 | 90.094 | 41.215 | 27 | 126.37 | 109.339 | 26 | 1.7% | -0.44 [-0.98, 0.11] | |
| Sugimoto et al. 2008 | 63.0147 | 39.1713 | 20 | 97.0767 | 83.4519 | 10 | 1.5% | -0.58 [-1.35, 0.20] | |
| Tan et al. 2009 | 39.3416 | 27.0793 | 41 | 39.001 | 36.9573 | 21 | 1.7% | 0.01 [-0.52, 0.54] | |
| Tao et al. 2013a | 82.65 | 57.37 | 24 | 96.29 | 78.97 | 23 | 1.6% | -0.19 [-0.77, 0.38] | |
| Tao et al. 2013b | 82.65 | 57.37 | 24 | 128.43 | 89.94 | 24 | 1.6% | -0.60 [-1.18, -0.02] | |
| Testa R et al. 1989 | 91.34 | 37.57 | 10 | 89.95 | 45.97 | 20 | 1.5% | 0.03 [-0.73, 0.79] | |
| Thomsen et al. 2016a | 81.7488 | 18.7341 | 23 | 103.889 | 23.8434 | 39 | 1.7% | -0.99 [-1.53, -0.44] | |
| Thomsen et al. 2016b | 81.7488 | 18.7341 | 23 | 105.592 | 20.4372 | 44 | 1.7% | -1.19 [-1.73, -0.64] | |
| Tsai et al. 2019 | 29.8 | 12.3 | 36 | 32.3 | 15.5 | 18 | 1.6% | -0.18 [-0.75, 0.38] | |
| Wang et al. 2017 | 44.3 | 4.2 | 235 | 57.9 | 8.9 | 131 | 1.8% | -2.16 [-2.42, -1.89] | |
| Yoon et al. 2019 | 170.31 | 103.889 | 175 | 149.87 | 71.5302 | 46 | 1.8% | 0.21 [-0.12, 0.53] | |
| Zhang et al. 2013 | 80.7951 | 55.7254 | 40 | 106.035 | 48.7087 | 27 | 1.7% | -0.47 [-0.97, 0.02] | |
| Zhang et al. 2013* | 69.6227 | 51.3144 | 25 | 178.826 | 60.0683 | 26 | 1.6% | -1.92 [-2.59, -1.25] | |
| Zhang et al. 2014 | 87.9481 | 56.3896 | 29 | 101.982 | 52.183 | 22 | 1.7% | -0.25 [-0.81, 0.30] | |
| Zhang et al. 2017 | 112.745 | 54.3289 | 25 | 155.834 | 51.7742 | 30 | 1.7% | -0.80 [-1.36, -0.25] | |
| Zhang et al. 2018 | 106.955 | 43.94 | 48 | 144.764 | 55.8617 | 38 | 1.7% | -0.76 [-1.20, -0.32] | |
| Zheng et al. 2013 | 94.5221 | 64.37 | 20 | 106.273 | 47.17 | 16 | 1.6% | -0.20 [-0.86, 0.46] | |
| Zheng et al. 2014 | 89.9237 | 58.9273 | 31 | 104.23 | 49.0493 | 24 | 1.7% | -0.26 [-0.79, 0.28] | |
| Zheng et al. 2017a | 67.7834 | 39.8525 | 14 | 143.912 | 75.1067 | 8 | 1.4% | -1.34 [-2.31, -0.36] | |
| Zheng et al. 2017b | 67.7834 | 39.8525 | 14 | 54.4992 | 26.909 | 11 | 1.5% | 0.37 [-0.43, 1.17] | |
| Zhou et al. 2016a | 63.2191 | 26.0234 | 52 | 180.886 | 78.581 | 19 | 1.6% | -2.53 [-3.21, -1.85] | |
| Zhou et al. 2016b | 63.2191 | 26.0234 | 52 | 108.675 | 32.7336 | 21 | 1.6% | -1.60 [-2.17, -1.03] | |
| **Total (95% CI)** | | | **2558** | | | **1771** | **100.0%** | **-0.91 [-1.14, -0.69]** | |

Heterogeneity: Tau² = 0.71; Chi² = 629.84, df = 62 (P < 0.00001); I² = 90%
Test for overall effect: Z = 7.96 (P < 0.00001)

**Fig 2. Forest plot for estimating the association of circulatory ammonia and HE.** A random-effect model was used to compare the standardized mean difference of ammonia between groups. The below funnel plot represents potential publication bias in the study. HE: Hepatic encephalopathy.

89%; Fig 5A). Furthermore, 5 studies were included to compare the circulating levels of tumor necrosis factor-alpha (TNFα) between both groups (313 participants). The average levels of TNFα were significantly increased in HE patients compared to control (P < 0.00001; I² = 55%; Fig 5B).

a

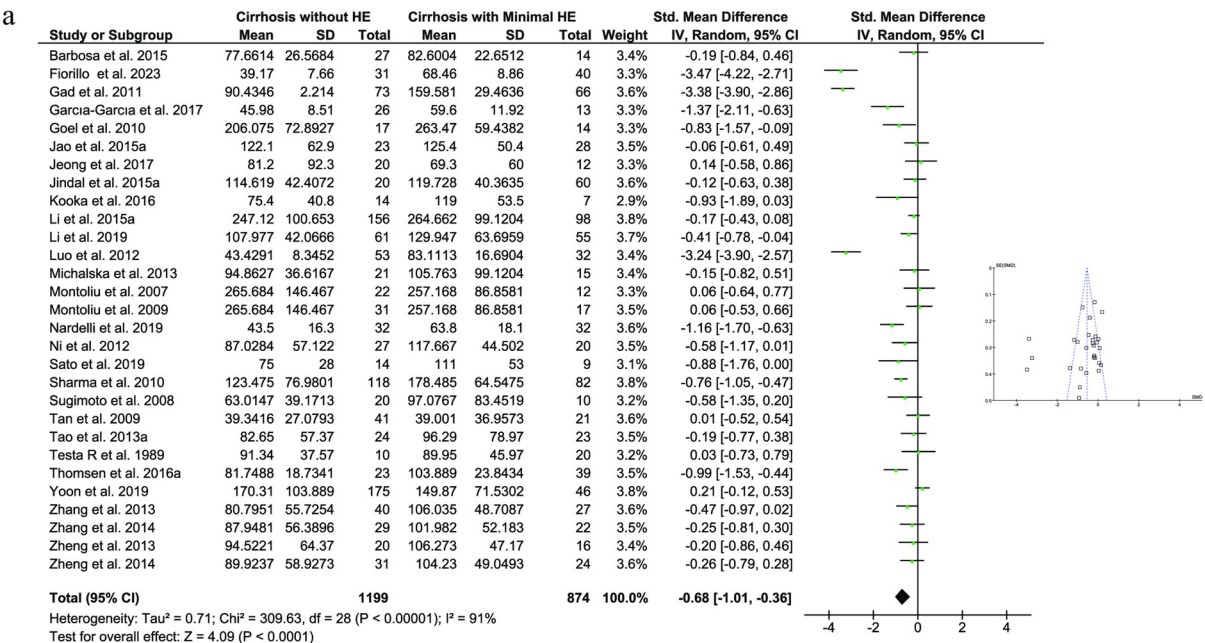

b

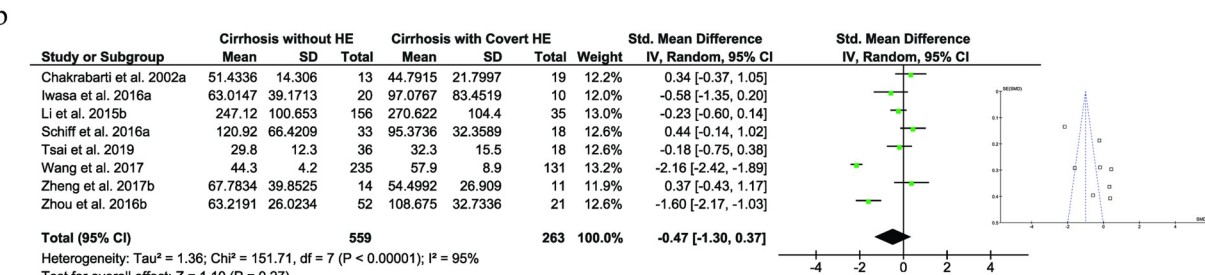

c

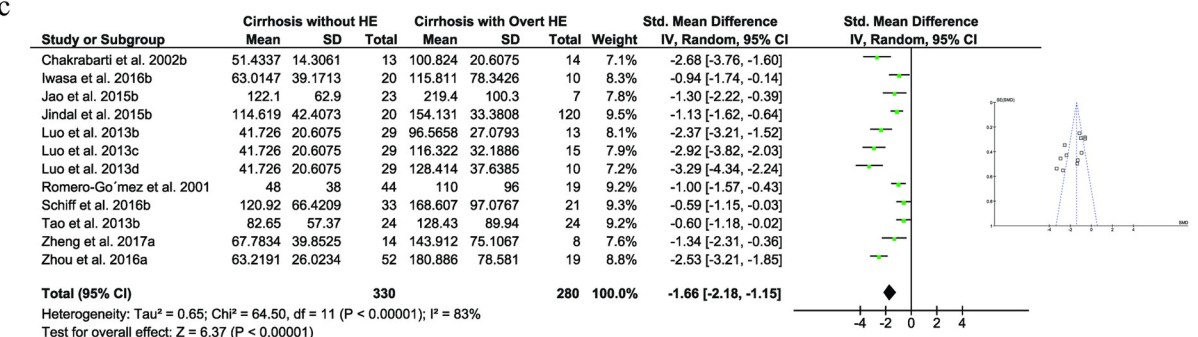

**Fig 3. Forest plot for estimating the association of circulatory ammonia and type of HE.** A random-effect model was used to compare the standardized mean difference of ammonia between groups. Comparing average levels of ammonia between cirrhotic patients without HE and cirrhotic patients with minimal HE (a), cirrhotic patients with covert HE (b), and cirrhotic patients with overt HE (c). Funnel plots represent potential publication bias. HE: Hepatic encephalopathy.

## Discussion

The pathophysiology of HE is not fully understood, and its prognosis is not very well in cirrhosis. Moreover, the association between elevated blood ammonia levels and the severity of HE remains controversial, despite reports suggesting these levels are a major predictor of hospitalization and mortality in individuals with advanced liver disease and liver failure [115–117]. This

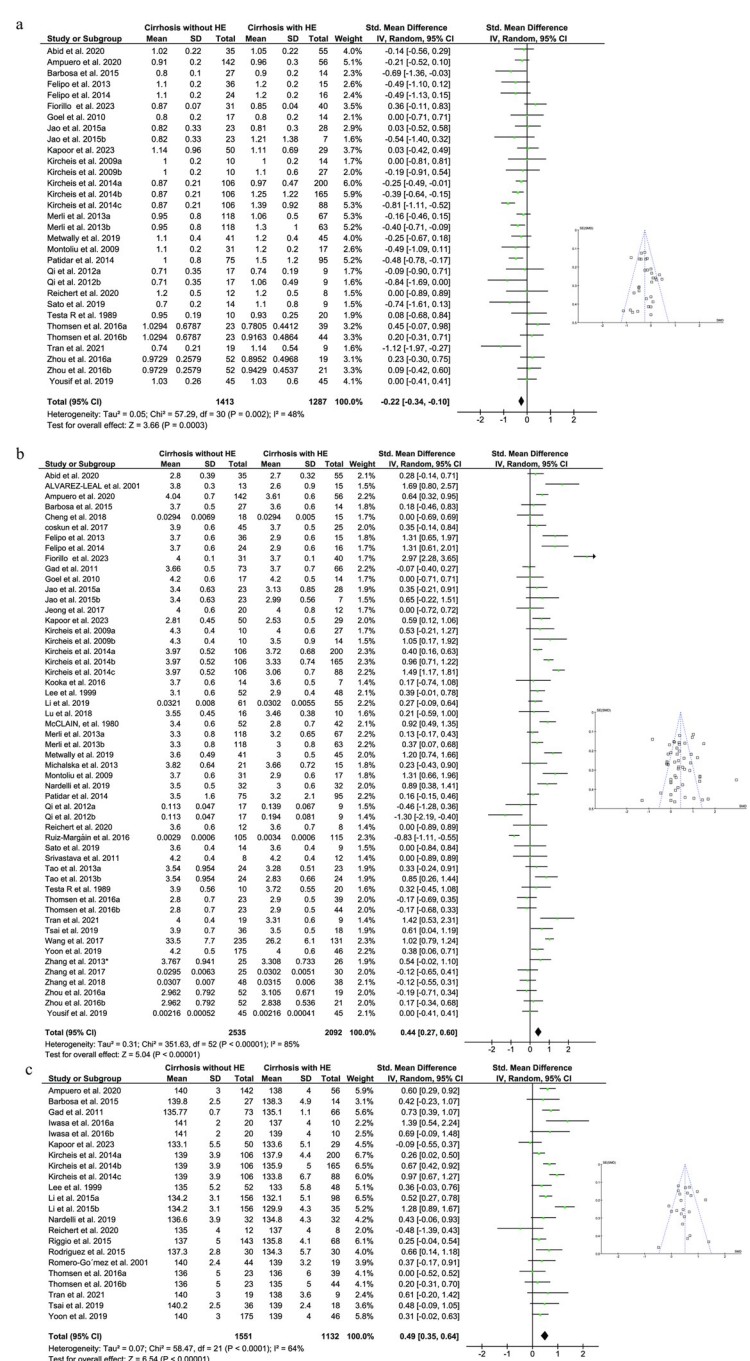

**Fig 4.** Forest plot for estimating the standardized mean difference of circulatory creatinine (a), albumin (b) and sodium (c) between control and HE groups. A random-effect model was used in the meta-analysis. Funnel plots represent potential publication bias.

meta-analysis confirmed elevated levels of blood ammonia are associated with the development of HE in patients with cirrhosis. Surprisingly, the ammonia levels were raised in cirrhotic patients independent of the type of HE. However, this observation was more reliable in individuals with OHE as they showed lower heterogeneity. This could indicate that hyperammonemia mediated the progression of HE from subtle cognitive changes to severe personality changes

a

| Study or Subgroup | Cirrhosis without HE | | | Cirrhosis with HE | | | Weight | Std. Mean Difference IV, Random, 95% CI | Std. Mean Difference IV, Random, 95% CI |
|---|---|---|---|---|---|---|---|---|---|
| | Mean | SD | Total | Mean | SD | Total | | | |
| Felipo et al. 2014 | 2.3 | 0.3 | 24 | 4.4 | 0.5 | 16 | 6.6% | -5.26 [-6.63, -3.89] | |
| García-García et al. 2017 | 2.4 | 0.2 | 26 | 4.2 | 0.6 | 13 | 6.9% | -4.65 [-5.93, -3.37] | |
| Luo et al. 2012 | 24.6 | 18.5 | 53 | 46.4 | 15.2 | 32 | 9.2% | -1.25 [-1.72, -0.77] | |
| Luo et al. 2013a | 32.5 | 11.7 | 29 | 44.3 | 13.5 | 17 | 8.8% | -0.94 [-1.57, -0.31] | |
| Luo et al. 2013b | 32.5 | 11.7 | 29 | 52.9 | 16.8 | 13 | 8.6% | -1.49 [-2.22, -0.75] | |
| Luo et al. 2013c | 32.5 | 11.7 | 29 | 65.9 | 15.4 | 15 | 8.3% | -2.51 [-3.34, -1.68] | |
| Luo et al. 2013d | 32.5 | 11.7 | 29 | 82.2 | 29.5 | 10 | 7.9% | -2.74 [-3.71, -1.78] | |
| Montoliu et al. 2009 | 6 | 4 | 31 | 15 | 2 | 17 | 8.4% | -2.57 [-3.37, -1.77] | |
| Rodriguez et al. 2015 | 27.3 | 69.3 | 30 | 57 | 96 | 30 | 9.1% | -0.35 [-0.86, 0.16] | |
| Srivastava et al. 2011 | 16.5 | 6 | 8 | 19.6 | 6.7 | 12 | 8.1% | -0.46 [-1.37, 0.45] | |
| Tsai et al. 2015 | 1.98 | 0.69 | 67 | 3.17 | 1.63 | 27 | 9.2% | -1.13 [-1.61, -0.65] | |
| Tsai et al. 2019 | 1.9 | 0.7 | 36 | 3.3 | 1.7 | 18 | 8.9% | -1.22 [-1.84, -0.61] | |
| **Total (95% CI)** | | | **391** | | | **220** | **100.0%** | **-1.92 [-2.54, -1.30]** | |

Heterogeneity: Tau² = 1.03; Chi² = 102.52, df = 11 (P < 0.00001); I² = 89%
Test for overall effect: Z = 6.05 (P < 0.00001)

b

| Study or Subgroup | Cirrhosis without HE | | | Cirrhosis with HE | | | Weight | Std. Mean Difference IV, Random, 95% CI | Std. Mean Difference IV, Random, 95% CI |
|---|---|---|---|---|---|---|---|---|---|
| | Mean | SD | Total | Mean | SD | Total | | | |
| Luo et al. 2012 | 39.2 | 12.7 | 53 | 53.9 | 18.5 | 32 | 24.2% | -0.96 [-1.43, -0.50] | |
| Rodriguez et al. 2015 | 1.5 | 6.4 | 30 | 22.6 | 27.6 | 30 | 21.3% | -1.04 [-1.58, -0.50] | |
| Srivastava et al. 2011 | 12.4 | 3.4 | 8 | 22.2 | 6.8 | 12 | 9.5% | -1.64 [-2.70, -0.58] | |
| Tsai et al. 2015 | 41.38 | 36.82 | 67 | 58.1 | 74.16 | 27 | 24.7% | -0.33 [-0.78, 0.12] | |
| Tsai et al. 2019 | 35.7 | 33.2 | 36 | 60.7 | 77.1 | 18 | 20.3% | -0.48 [-1.05, 0.10] | |
| **Total (95% CI)** | | | **194** | | | **119** | **100.0%** | **-0.79 [-1.17, -0.41]** | |

Heterogeneity: Tau² = 0.10; Chi² = 8.86, df = 4 (P = 0.06); I² = 55%
Test for overall effect: Z = 4.09 (P < 0.0001)

**Fig 5.** Forest plot for estimating the standardized mean difference of IL-6 (a), and TNFα (b) between control and HE groups. Funnel plots represent potential publication bias.

and gross disorientation that has been found in patients with OHE [118, 119]. We must emphasize that our data confirmed the notion that patients with HE had significantly higher MELD score, Child-Pugh score, bilirubin, ALT, AST, and GGT (Supplementary results in S1 File). Due to poor prognosis associated with HE, we proceeded to our meta-analysis centered on multiple laboratory data to identify potential predictors for that. This study is the first systematic review and meta-analysis comparing levels of circulating ammonia, creatinine, albumin and sodium between cirrhotic patients with HE and cirrhotic volunteers without HE. Another intriguing observation is that patients with HE exhibited elevated average circulating creatinine levels. Studies have demonstrated a correlation between abnormal blood creatinine levels and the severity of HE, particularly in patients with hepatitis C [120]. The raised in blood creatinine is

associated with kidney injury and mortality in cirrhotic patients [121]. Moreover, a retrospective study has demonstrated a correlation between higher circulating creatinine levels and hospital mortality in cirrhosis patients with HE [122]. The mechanism underlying kidney damage following cirrhosis is likely due to hemodynamic impairments. A series of these impairments, including portal hypertension, arterial vasodilation, ascites, hypotension, increased cardiac output, hypovolemia, activation of the renin-angiotensin-aldosterone system, and renal vasoconstriction, are considered to contribute to the development of kidney injury and renal dysfunction in the context of cirrhosis [123–125]. The meta-analysis also showed that cirrhotic patients with HE had decreased circulating albumin and sodium levels compared to individuals without HE. The results of multiple investigations showed that albumin infusion improved survival and decreased the mortality risk and progression of OHE in cirrhotic patients [126–128]. The purpose of albumin infusion in cirrhotic patients with HE is to promote plasma expansion, bind to toxic blood components, enhance antioxidant capacity, and have anti-inflammatory properties [129–132]. Consequently, hypoalbuminemia, which is induced by the depletion of hepatocyte mass, may function as a clinical indicator of hepatic encephalopathy (HE) and its severity in cirrhosis. In comparison to the control group, we noted a decrease of approximately 1.8 mEq/L in the average blood sodium levels in patients with HE. Notwithstanding, the blood sodium concentrations in both groups remained within the normal range of 135–145 mEq/L. It's important to note that hyponatremia occurs when blood sodium levels fall below 135 mEq/L. In individuals with cirrhosis, hyponatremia is associated with increased morbidity and mortality as well as a higher grade of HE [133]. Additionally, cirrhotic patients who had blood sodium levels below 135 mEq/L were more likely to have HE [134]. In more than 75% of cirrhotic individuals with HE, hyponatremia has been seen, and more importantly, the reduction of blood sodium concentration was associated with the main consequences of cirrhosis, including ascites, coagulopathy, and spontaneous bacterial peritonitis [135]. Even when blood sodium reduction is within normal limits, its value should still be closely monitored in the circulation of patients with cirrhosis in terms of the potential occurrence of HE. The results of the current meta-analysis also revealed exacerbation of systemic inflammation in cirrhotic patients with HE, as evidenced by significantly elevated circulating levels of TNFα and IL-6. There is increasing evidence that in addition to ammonia, brain dysfunction following cirrhosis is also caused by systemic inflammatory response syndrome [136, 137]. Systemic inflammation is also associated with severity of MHE and progression to OHE [25, 64, 138]. Nevertheless, other research groups have demonstrated that cirrhosis does not necessarily lead to HE, and cognitive deficits are not solely due to inflammation [30, 31]. There is greater agreement among studies regarding the synergistic roles of inflammation and ammonia in the development of HE, and its severity in cirrhosis [31, 138–140]. This meta-analysis clearly demonstrated that a substantial enhancement of these two factors in cirrhotic individuals with HE in comparison to cirrhotic volunteers without HE. While several factors appear to contribute to the pathophysiology of, HE due to multifactorial nature of disease, rapid identification of blood diagnostic laboratory parameters that revealed in this study may enable physicians and researchers to stop the course of disease more quickly. The primary limitation of this study is the significant heterogeneity observed in several examined parameters. This heterogeneity primarily stems from methodological variations across the papers, including disparities in reported units that necessitated conversion, differences in sample sizes, and geographic locations.

## Conclusion

This study has found a strong link between ammonia levels in the blood and HE in cirrhotic patients, particularly in OHE. Additionally, it is essential to closely monitor follow-up levels of

creatinine, albumin, sodium, and systemic inflammation, as these may serve as significant prognostic indicators for hospital mortality and the progression of HE in cirrhosis patients. It is worth noting that because HE is a complex disease with multiple factors at play, relying solely on ammonia-scavenging strategies for treatment is not recommended. Instead, exploring novel approaches that target inflammation, creatinine, albumin, sodium, and ammonia levels in the bloodstream may be more beneficial.

## Supporting information

**S1 File.**
(DOCX)

**S1 Fig. Forest plot for estimating the standardized mean difference of Child-Pugh, and MELD scores between control and HE groups.**
(TIF)

**S2 Fig. Forest plot for estimating the standardized mean difference of circulatory PT and values of INR between control and HE groups.**
(TIF)

**S3 Fig. Forest plot for estimating the standardized mean difference of circulatory total bilirubin between control and HE groups.**
(TIF)

**S4 Fig. Forest plot for estimating the standardized mean difference of circulatory ALT, AST, and GGT between control and HE groups.**
(TIF)

**S5 Fig. Forest plot for estimating the standardized mean difference of hemoglobin, platelets and white blood cells between control and HE groups.**
(TIF)

## Author Contributions

**Conceptualization:** Ali Sepehrinezhad, Negin Ghiyasi Moghaddam, Navidreza Shayan.

**Data curation:** Ali Sepehrinezhad, Navidreza Shayan, Sajad  Sahab Negah.

**Formal analysis:** Negin Ghiyasi Moghaddam, Sajad  Sahab Negah.

**Investigation:** Ali Sepehrinezhad.

**Methodology:** Ali Sepehrinezhad, Negin Ghiyasi Moghaddam, Navidreza Shayan, Sajad Sahab Negah.

**Project administration:** Ali Sepehrinezhad, Sajad  Sahab Negah.

**Supervision:** Sajad  Sahab Negah.

**Validation:** Sajad  Sahab Negah.

**Visualization:** Ali Sepehrinezhad.

**Writing – original draft:** Ali Sepehrinezhad, Negin Ghiyasi Moghaddam, Navidreza Shayan.

**Writing – review & editing:** Sajad  Sahab Negah.

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
