## [Decision Letter · Decision Letter 0]

10 May 2024

PONE-D-24-09594Correlation of Ammonia and Blood Laboratory Parameters with Hepatic Encephalopathy: A Systematic Review and Meta-AnalysisPLOS ONE

Dear Dr. Sahab Negah,

Thank you for submitting your manuscript to PLOS ONE. After careful consideration, we feel that it has merit but does not fully meet PLOS ONE’s publication criteria as it currently stands. Therefore, we invite you to submit a revised version of the manuscript that addresses the points raised during the review process.

**In particular reviewer 2 raised serious concerns regarding several methodological aspects. Those concerns need to be thoroughly addressed and the required changes and adaptations need to be made to the manuscript. This likely requires additional work and potentially re-analysis of the data. All changes must be clearly highlighted in the paper and a complete and detailed point by point reply explaining how the criticism has been addressed needs to be provided if the authors opt for revision of their paper.** Please submit your revised manuscript by Jun 24 2024 11:59PM. If you will need more time than this to complete your revisions, please reply to this message or contact the journal office at plosone@plos.org. Please include the following items when submitting your revised manuscript:A rebuttal letter that responds to each point raised by the academic editor and reviewer(s). You should upload this letter as a separate file labeled 'Response to Reviewers'.A marked-up copy of your manuscript that highlights changes made to the original version. You should upload this as a separate file labeled 'Revised Manuscript with Track Changes'.An unmarked version of your revised paper without tracked changes. You should upload this as a separate file labeled 'Manuscript'.

We look forward to receiving your revised manuscript.

Kind regards,

Peter Starkel, M.D., Ph.D.

Academic Editor

PLOS ONE

Journal Requirements:

Additional Editor Comments:

In particular reviewer 2 raised serious concerns regarding several methodological aspects. Those concerns need to be thoroughly addressed and the required changes and adaptations need to be made to the manuscript. This likely requires additional work and potentially re-analysis of the data. All changes must be clearly highlighted in the paper and a complete and detailed point by point reply explaining how the criticism has been addressed needs to be provided if the authors opt for revision of their paper.

Reviewers' comments:

Reviewer's Responses to Questions

**Comments to the Author**

1. Is the manuscript technically sound, and do the data support the conclusions?

Reviewer #1: Yes

Reviewer #2: No

2. Has the statistical analysis been performed appropriately and rigorously? 

Reviewer #1: Yes

Reviewer #2: No

3. Have the authors made all data underlying the findings in their manuscript fully available?

Reviewer #1: Yes

Reviewer #2: No

4. Is the manuscript presented in an intelligible fashion and written in standard English?

Reviewer #1: Yes

Reviewer #2: No

5. Review Comments to the Author

**Reviewer #1:** Hepatic encephalopathy seen in liver cirrhosis is thought to involve multiple factors, with ammonia believed to play a central role. However, there are reports suggesting that the pathophysiology of hepatic encephalopathy cannot be accurately assessed based solely on blood ammonia levels. The authors conducted a meta-analysis of 79 studies, investigating not only the correlation between hepatic encephalopathy and blood ammonia levels in liver cirrhosis but also its association with creatinine, albumin, sodium, IL-6, and TNF-α concentrations. As a result, cases with hepatic encephalopathy showed significantly elevated levels of ammonia, creatinine, IL-6, and TNF-α, while albumin and sodium levels were significantly lower. Therefore, the authors concluded that relying solely on targeting ammonia in the treatment of hepatic encephalopathy might not be sufficient and suggested a new strategy involving addressing inflammation, creatinine, albumin, and sodium concentrations. This conclusion is supported by a comprehensive examination of numerous cases using appropriate methods. However, there are several points that warrant further discussion.

The authors also note significant variability in ammonia levels. One possible cause is the variation in ammonia reference values among studies. In multicenter studies involving ammonia levels, some use ratios to the upper limit of the reference value rather than raw data. It would be beneficial to address how this aspect is evaluated.

Blood ammonia levels are not necessarily an appropriate indicator for evaluating hepatic encephalopathy, partly because many of the examined ammonia values are from venous blood samples. Venous blood ammonia, being detoxified through the glutamine synthesis pathway in skeletal muscles, tends to be lower than arterial blood levels. However, since the ammonia entering the brain is transported by arterial blood, venous blood ammonia levels may not accurately reflect the concentration of ammonia flowing into the brain. This issue also warrants discussion in the analysis.

**Reviewer #2**: In this paper, Sepehrinezhad A and coll. attempted to synthetize the correlation of blood ammonia with hepatic encephalopathy in a meta-analysis. I have major concerns about the methodology of this paper, particularly regarding the literature search, the criteria for inclusion and exclusion of studies which are missing and the statistical analyses. Key search terms must be noted in the main text and must be combined within each database (not done), abstracts from liver congresses are usually screened (not done), literature search must be updated in April 2024 and some studies were not identified. In order to reduce risk of bias, strict criteria for inclusion and exclusion of studies must be clearly defined prior to the literature search. Also, aims of the study and endpoints are missing in the text. The selection of the studies for inclusion in the metaanalysis includes usually 4 processes which is mandatory to preserve: identification, screening, eligibility (missing) and inclusion (Fig 1). Regarding statistical analysis, I2 alone is not enough to assess heterogeneity between studies. Moreover, in cases of moderate or high heterogeneity, the methodological section of each study is usually re-reviewed to determine whether any discrepancy could be identified, and sensitivity analyses excluding the discrepant study is classically performed (not done). Therefore, because the methodology and statistical analyses were not complete and rigorous, I recommend rejecting this article.

6. PLOS authors have the option to publish the peer review history of their article (what does this mean?). If published, this will include your full peer review and any attached files.

Reviewer #1: No

Reviewer #2: No

---

## [Author Response · Author response to Decision Letter 0]

5 Jul 2024

To: 5th, July, 2024

To: PLOS ONE, Editor-in-Chief

Dear Editor-in-Chief

We would like to express our appreciation to the reviewer for his/her insightful comments, which have significantly helped us improve our manuscript. We have incorporated your feedback into our revisions, and we believe that your comments have helped to clarify and strengthen our paper. Please find our responses to your specific comments below (in blue), and note that the revised text in the manuscript is highlighted in blue for your convenience. 

Reviewer #1:

Hepatic encephalopathy seen in liver cirrhosis is thought to involve multiple factors, with ammonia believed to play a central role. However, there are reports suggesting that the pathophysiology of hepatic encephalopathy cannot be accurately assessed based solely on blood ammonia levels. The authors conducted a meta-analysis of 79 studies, investigating not only the correlation between hepatic encephalopathy and blood ammonia levels in liver cirrhosis but also its association with creatinine, albumin, sodium, IL-6, and TNF-α concentrations. As a result, cases with hepatic encephalopathy showed significantly elevated levels of ammonia, creatinine, IL-6, and TNF-α, while albumin and sodium levels were significantly lower. Therefore, the authors concluded that relying solely on targeting ammonia in the treatment of hepatic encephalopathy might not be sufficient and suggested a new strategy involving addressing inflammation, creatinine, albumin, and sodium concentrations. This conclusion is supported by a comprehensive examination of numerous cases using appropriate methods. However, there are several points that warrant further discussion. The authors also note significant variability in ammonia levels. One possible cause is the variation in ammonia reference values among studies. In multicenter studies involving ammonia levels, some use ratios to the upper limit of the reference value rather than raw data. It would be beneficial to address how this aspect is evaluated. Blood ammonia levels are not necessarily an appropriate indicator for evaluating hepatic encephalopathy, partly because many of the examined ammonia values are from venous blood samples. Venous blood ammonia, being detoxified through the glutamine synthesis pathway in skeletal muscles, tends to be lower than arterial blood levels. However, since the ammonia entering the brain is transported by arterial blood, venous blood ammonia levels may not accurately reflect the concentration of ammonia flowing into the brain. This issue also warrants discussion in the analysis.

Our response: Thank you for your valuable comment. Based on various clinical studies, it has been observed that both venous and arterial ammonia levels have strong correlations with the severity of hepatic encephalopathy. Some studies even suggest a venous sample is adequate for measuring ammonia levels. Additionally, since arterial ammonia levels are higher than venous levels, it is believed that the venous level can accurately represent the level that may lead to brain injury, as arterial ammonia reaches the brain more quickly. Therefore, venous ammonia levels can be considered as reliable biomarkers. Thus, it appears that our systematic review and meta-analysis can overlook the kind of arterial and venous sampling used for ammonia measurement.

Mehmood, M. A., T. Waseem, F. Z. Ahmad, and M. A. Humayun. "Measuring partial pressure of ammonia in arterial or venous blood vs total ammonia levels in hepatic encephalopathy." J. Gastroenterol. Hepatol 2 (2013): 602.

Nicolao, Francesca, Cesare Efrati, Andrea Masini, Manuela Merli, Adolfo Francesco Attili, and Oliviero Riggio. "Role of determination of partial pressure of ammonia in cirrhotic patients with and without hepatic encephalopathy." Journal of hepatology 38, no. 4 (2003): 441-446.

Ong JP, Aggarwal A, Krieger D, Easley KA, Karafa MT, Van Lente F, Arroliga AC, Mullen KD. Correlation between ammonia levels and the severity of hepatic encephalopathy. Am J Med. 2003 Feb 15;114(3):188-93. doi: 10.1016/s0002-9343(02)01477-8. PMID: 12637132.

K.N, Sricharan and Shifali Prabhaker. “Arterial vs Venous Ammonia Levels in Correlation with severity of Hepatic Encephalopathy.” International Journal of Contemporary Medical Research [IJCMR] (2019): n. pag.

Reviewer #2:

In this paper, Sepehrinezhad A and coll. attempted to synthetize the correlation of blood ammonia with hepatic encephalopathy in a meta-analysis. I have major concerns about the methodology of this paper, particularly regarding the literature search, the criteria for inclusion and exclusion of studies which are missing and the statistical analyses. Key search terms must be noted in the main text and must be combined within each database (not done), abstracts from liver congresses are usually screened (not done), literature search must be updated in April 2024 and some studies were not identified. In order to reduce risk of bias, strict criteria for inclusion and exclusion of studies must be clearly defined prior to the literature search. Also, aims of the study and endpoints are missing in the text. The selection of the studies for inclusion in the metaanalysis includes usually 4 processes which is mandatory to preserve: identification, screening, eligibility (missing) and inclusion (Fig 1). Regarding statistical analysis, I2 alone is not enough to assess heterogeneity between studies. Moreover, in cases of moderate or high heterogeneity, the methodological section of each study is usually re-reviewed to determine whether any discrepancy could be identified, and sensitivity analyses excluding the discrepant study is classically performed (not done). Therefore, because the methodology and statistical analyses were not complete and rigorous, I recommend rejecting this article.

Our response: I appreciate your attention to detail and thoughtful critique of the methodology used in this study. As you we have suggested, expanded our literature search to include publications up to June 2024 and have incorporated our key search terms into the main body of the revised manuscript. The primary keyword search pattern for each database is now available in Supplementary Table 2. For the following comment “abstracts from liver congresses are usually screened (not done)”, as we mentioned in the Prisma flowchart, we excluded abstracts from the conference because they didn’t have enough data for concluding as well as there weren’t inclusion criteria for the conference paper. 

For the comment “In order to reduce risk of bias, strict criteria for inclusion and exclusion of studies must be clearly defined prior to the literature search”, I would like to confirm that we have implemented these measures. We established general exclusion criteria, which included non-English papers, abstracts, and reviews. Additionally, we only included studies that met the following specific criteria: i) studies that compared cirrhosis with hepatic encephalopathy, ii) the use of a valid evaluation method, iii) a valid and clear definition of cases, and iv) the measurement of outcomes using a reproducible and reliable method.

To further minimize the risk of bias, we conducted an appraisal utilizing the Joanna Briggs Institute (JBI) critical appraisal checklist for case-control studies. This approach shows that an appraisal checklist can effectively mitigate the risk of bias in studies with poor methodology.

In the comment, "The selection of the studies for inclusion in the meta-analysis involves four mandatory processes: identification, screening, eligibility (missing), and inclusion (Fig 1)," the eligibility of studies was assessed at three levels: general criteria, PICO criteria, and outcome measurements. For instance, we considered Population (i.e., cirrhosis and hepatic encephalopathy), Comparison (HE as the case and cirrhosis as the control), and Outcome (e.g., serum biomarkers). Given that our manuscript focused on evaluating observational studies, we intentionally refrained from providing a specific definition of intervention; instead, we exclusively included observational studies. 

Regarding the comment, "I2 alone is not sufficient to assess heterogeneity between studies in statistical analysis. In cases of moderate or high heterogeneity, the methodological section of each study is usually re-reviewed to identify discrepancies, and sensitivity analyses excluding the discrepant study are typically performed (not done)," we precisely reviewed the methodology of each study and focused on the assessment methods, such as the evaluation of inflammatory markers using the Eliza method. Additionally, we only included similar studies in the sub-analysis that differed in one variable.

I believe the major comments on the methodology were due to the concise nature of this part. The manuscript's word limitation was the main challenge we encountered. Following the aforementioned details, we have provided a comprehensive description of the comments in the supplementary document.

Sincerely yours

Dr. Sajad Sahab Negah, PhD Department of Neuroscience, Faculty of Medicine, Mashhad University of Medical Sciences, Pardis Campus, Azadi Square, Kalantari Blvd., Mashhad, Iran. Tel: +98-51-38002473; Email: sahabnegahs@mums.ac.ir

---

## [Editor Report · Decision Letter 1]

15 Jul 2024

Correlation of Ammonia and Blood Laboratory Parameters with Hepatic Encephalopathy: A Systematic Review and Meta-Analysis

PONE-D-24-09594R1

Dear Dr. Sahab Negah,

We’re pleased to inform you that your manuscript has been judged scientifically suitable for publication and will be formally accepted for publication once it meets all outstanding technical requirements.

Kind regards,

Peter Starkel, M.D., Ph.D.

Academic Editor

PLOS ONE

Additional Editor Comments (optional):

Data have been updated and sufficient clarifications concerning the methods used have been provided.
---

## [Editor Report · Acceptance letter]

23 Jul 2024

PONE-D-24-09594R1 

PLOS ONE

Dear Dr. Sahab Negah, 

I'm pleased to inform you that your manuscript has been deemed suitable for publication in PLOS ONE. Congratulations! Your manuscript is now being handed over to our production team.

Kind regards, 

on behalf of

Dr Peter Starkel 

Academic Editor

PLOS ONE